# Fluorine-Terminated Liquid Polybutadiene: A Novel Approach to Enhancing Oil Resistance and Thermal Stability in Natural Rubber

**DOI:** 10.3390/ijms26073410

**Published:** 2025-04-05

**Authors:** Xue Luo, Mengyan Li, Guliang Fu, Rentong Yu, Jianhe Liao

**Affiliations:** School of Materials Science & Engineering, Hainan University, Haikou 570228, China; 19834407278@163.com (X.L.); limengyan133xx1539@163.com (M.L.); 13907504371@163.com (G.F.)

**Keywords:** natural rubber, polybutadiene, oil resistance, fluorination modification

## Abstract

Natural rubber (NR) has long been plagued by inferior oil resistance and poor thermal degradation at high temperatures. Despite these limitations, NR remains the most widely used elastomer to date. Fluorine-containing compounds have demonstrated excellent oil resistance and thermal stability. However, they generally exhibit poor compatibility with non-polar polymers. After blending, macroscopic phase separation cannot be easily suppressed, leading to the deterioration of the material’s properties. In this study, fluorination modification was performed using hydroxyl-terminated polybutadiene, and the resulting fluorine-modified polybutadiene (3F-PBu-3F) was incorporated into natural rubber. Following sulfur curing, homogeneous phase morphologies were observed in all vulcanizates, which significantly differed from those of previously reported NR/polybutadiene vulcanizates. Additionally, the oil resistance and thermal stability of the NR/3F-PBu-3F vulcanizates were effectively enhanced.

## 1. Introduction

Natural rubber (NR), a renewable elastomer derived from the para rubber tree (*Hevea brasiliensis*), has long been the most widely used elastomer [1]. Compared to synthetic rubbers, NR remains highly competitive and irreplaceable due to its exceptional comprehensive properties, including high tensile strength, superior elasticity, low heat build-up, and cost-effectiveness [2,3]. In 2021, 13.77 million tonnes of natural rubber was utilized as the primary material in applications such as tires, conveyor belts, vibration dampers, and more [4]. Most NR products are sulfur-vulcanized for practical engineering applications. However, certain limitations hinder the broader application of NR in sulfur-cured systems. Firstly, a heterogeneous network often forms due to the poor and inconsistent dispersion of sulfur and accelerators within the natural rubber matrix [5]. This results in defects such as dangling chains, topological constraints, and sol chains distributed throughout the NR network [6]. Secondly, NR struggles to perform effectively in oil-containing environments because of its non-polar molecular structure. To address this, significant efforts have been made through molecular design strategies, including reducing the double bond content by optimizing the vulcanization system, introducing polar groups via chemical reactions, blending with partially polarized polymers, and impeding oil diffusion by incorporating nanofillers [7,8,9,10,11,12]. Thirdly, the poor thermal stability of NR, caused by chain scission of the carbon backbone, unstable polysulfide crosslinks, and oxidation of unsaturated bonds, necessitates improvement through various physical and chemical modifications [13,14,15,16,17,18,19,20].

Fluorine has high electronegativity (3.98) and a small atomic radius (0.67 Å), which makes the carbon–fluorine bond highly stable, endowing fluorinated materials with unique chemical inertness [21]. The inertness not only provides fluorinated materials with excellent hydrophobicity and lipophobicity but also imparts exceptional solvent and corrosion resistance [22]. NR, composed of cis-polyisoprene, lacks functional groups to react with fluorine atoms, rendering the direct fluorination of natural rubber challenging [23]. Hydroxyl-terminated polybutadiene (OH-PBu-OH) is an elastic polymer with high chemical reactivity due to its terminal hydroxyl groups (-OH), which facilitate reactions with various chemical moieties [24]. Upon fluorination, the terminal fluorine groups in 3F-PBu-3F molecules enhance both chemical stability and solvent resistance, making 3F-PBu-3F an ideal modifier for NR. By blending 3F-PBu-3F with NR, the difficulties associated with direct fluorination are circumvented, while the hydrophobicity, chemical stability, environmental resistance, and aging resistance of natural rubber are significantly improved through synergistic chemical and physical interactions.

In this study, fluorine-terminated liquid polybutadiene (3F-PBu-3F) was synthesized and subsequently incorporated into natural rubber (NR). It was anticipated that 3F-PBu-3F was compatible with NR due to specific hydrogen bonding interactions (between the fluorine atoms of 3F-PBu-3F and the proteins in NR). Moreover, oil resistance, together with thermal stability, was expected to be improved due to the incorporation of a fluorine-containing compound. NR-based vulcanizates containing varying loadings of 3F-PBu-3F were prepared using sulfur as the curing agent. The fluorine-terminated liquid polybutadiene was characterized by ^1^H NMR and FT-IR spectroscopy. The morphologies of the NR-based vulcanizates were examined using scanning electron microscopy (SEM, Carl Zeiss AG, Baden-Württemberg, Germany) and further discussed in conjunction with FT-IR results. The oil resistance, mechanical properties, and thermal stability of the vulcanizates were systematically measured and investigated.

## 2. Results and Discussion

### 2.1. Synthesis of 4-((3-Oxo-3-(2,2,2-trifluoroethoxy)propyl)thio)butanoic Acid

We synthesized 4-(((3-Oxo-3-(2,2,2-trifluoroethoxy))propyl)thio)butyric acid by the reaction of 2,2,2-trifluoropropyl acrylate with mercaptopropionic acid. The ^1^H NMR spectrum of the product is shown in Figure 1. The characteristic signal at 5.53 ppm was attributed to the α-methylene proton (-CH_2_-) group of -CF_3_, indicating the successful introduction of the fluorinated group. At the same time, the product was also analyzed with FT-IR (Figure 2). There was an absorption peak of C-F stretching at 1280 cm^−1^, indicating the presence of the -CF_3_ group. In addition, 1710 cm^−1^ and 1750 cm^−1^ were the carbonyl (-C=O) stretching vibration peaks of -COOH from mercaptopropionic acid and -COO- from 2,2,2-trifluoroethyl acrylate, respectively. In addition, the strong peak at 3440 cm^−1^ was attributed to the O-H bond from -COOH, which confirmed the partial retention of the mercaptopropionic acid structural unit. The results of FT-IR and ^1^H NMR showed that 4-((3-Oxo-3-(2,2,2-trifluoroethoxy))propyl)thiobutyric acid was successfully obtained for the next reaction.

### 2.2. Structure Analysis of Fluoro-Terminated Polybutadiene (3F-PBu-3F)

As shown in Figure 3, *α*, *ω*-dihydroxy-terminated polybutadiene exhibits characteristic resonance signals at 5.4–4.9 ppm and 2.0 ppm, which are attributed to the olefinic protons (-CH=CH-) and methylene protons (-CH_2_-) adjacent to the double bonds of the oligomer, respectively. The terminal hydroxyl groups display weak resonance signals at 4.0–4.2 ppm. Upon the modification of the terminal hydroxyl groups with trifluoromethanesulfonyl chloride (CF_3_SO_2_Cl), the chemical shifts corresponding to the hydroxyl groups completely disappear. By comparing the peak area ratio of methylene to methine protons, it can be concluded that the trifluoromethanesulfonyl (-SO_2_CF_3_) group successfully reacted with the terminal groups of the *α*, *ω*-dihydroxy-terminated polybutadiene. This conclusion is further supported by the FT-IR results. In Figure 4, the trans-1,4 band at 965 cm^−1^ and the cis-1,4 bands at 1639 cm^−1^, along with the band at 965 cm^−1^, are characteristic of polybutadiene [25]. Additionally, the 1.2 structure can be identified by the bands at 993 cm^−1^ and 910 cm^−1^ [26]. After terminal group modification, the band intensity at approximately 3200 cm^−1^, which indicates the presence of hydroxyl groups, is significantly weakened. Furthermore, a new sharp band attributed to fluorine atoms is observed in the 3F-PBu-3F spectrum. By combining these findings with the ^1^H-NMR analysis discussed above, it can be concluded that 3F-PBu-3F was successfully synthesized.

### 2.3. Morphologies of NR/3F-PBu-3F Vulcanizates

NR-based blends containing polybutadiene are widely used in the manufacturing of truck tire tread stock, engine mounts, high-capacity laminate (HCL) rotor bearings, and other applications [27,28]. Despite sharing non-polar characteristics, natural rubber and polybutadiene exhibit limited miscibility in blended systems, as evidenced by microphase separation [29,30]. Miscibility is influenced by factors such as solubility parameters, molecular weight, and viscosity at the mixing temperature [31]. The solubility parameters of natural rubber and polybutadiene are 16.2 and 16.4 J^1^/^2^/cm^3^/^2^, respectively [32]. In this study, the morphologies of NR/3F-PBu-3F vulcanizates were examined using SEM imaging. As shown in Figure 5, homogeneous morphologies are observed for all crosslinked materials, which significantly differs from the findings mentioned above. The appropriate molecular weight of 3F-PBu-3F and the presence of specific hydrogen bonding interactions likely contribute to the improved miscibility between the two polymers. The molecular weight of the liquid 3F-PBu-3F used in this study is relatively low. This reduction in molecular weight minimizes the loss of mixing entropy, thereby suppressing the tendency for phase separation, in accordance with Flory–Huggins thermodynamics. Consequently, the low molecular weight of 3F-PBu-3F facilitates its penetration into the interchain voids of the NR network, promoting physical entanglement and interfacial adhesion through enhanced chain mobility [33,34]. Additionally, natural rubber typically contains 0.5–3% (*w*/*w*) proteins, and the specific hydrogen bonding interactions between these proteins and the fluorine atoms of 3F-PBu-3F further enhance miscibility. This will be discussed in detail below based on an FT-IR analysis.

In Figure 6, the typical hydrocarbon structure of NR is observed in all vulcanizates. The bands at 3100 cm^−1^ and 1500–500 cm^−1^ are attributed to carbon–carbon and carbon–hydrogen bonds, respectively. The band at approximately 840 cm^−1^ is characteristic of the C-H out-of-plane bending of cis-polyisoprene. C=C stretching bands are detected at around 1640 cm^−1^. Additionally, bands at 1630 cm^−1^ (amide I) and 1541 cm^−1^ (amide II), associated with proteins, are observed in the NR-based composites [35]. A broad FTIR band at 3200–3450 cm^−1^, ascribed to the amino groups of proteins, is also detected. Furthermore, the band at approximately 1630 cm^−1^ shifts to 1643 cm^−1^, indicating specific hydrogen bonding interactions between the proteins and the fluorine atoms. As a result, the miscibility between the NR matrix and 3F-PBu-3F is significantly improved, leading to the formation of a homogeneous phase in the NR/3F-PBu-3F vulcanizates.

### 2.4. Crosslink Density and Oil Resistance of NR/3F-PBu-3F Vulcanizates

Both natural rubber (NR) and polybutadiene can be crosslinked using sulfur as a crosslinking agent [29,30]. As depicted in Figure 7, compared to that of NR, the crosslink density of the vulcanizate increases from 16.57 to 20.69 and to 20.60 with the addition of 1 phr and 2 phr of 3F-PBu-3F, respectively. However, as the content of 3F-PBu-3F further increases, the crosslink density of the vulcanizates decreases to 16.59 and then rises slightly. In terms of reactivity, the double bonds in NR are more reactive than those in polybutadiene due to the electron-donating effect of the methyl groups in NR, which increases the electron density of the double bonds. In this study, 3F-PBu-3F, a liquid oligomer, effectively dissolves sulfur and accelerators and diffuses efficiently within the blends during the curing process. Consequently, the crosslink density of NR/3F-PBu-3F vulcanizates increases when the content of 3F-PBu-3F is appropriately low. However, the vulcanization rate of natural rubber is significantly faster than that of polybutadiene when sulfur is used as the curing agent. Additionally, the vulcanization rate of polybutadiene slows considerably immediately after scorch [32]. The lower reactivity of polybutadiene explains the observed trend in crosslink density when the content of 3F-PBu-3F exceeds 2 phr.

The oil resistance of the NR/3F-PBu-3F vulcanizates was evaluated using the swelling degree, measured in accordance with ASTM D471. As shown in Figure 8, the swelling degree of all vulcanizates exhibits an upward trend over time. However, the rate of increase in swelling degree gradually slows as time progresses. Additionally, for each identical test duration, the swelling degree of vulcanizates containing 3F-PBu-3F is consistently lower than that of pure natural rubber. This is particularly evident for the vulcanizate with a 3F-PBu-3F content of 2 phr. After 7 days of swelling, its swelling degree is 68%, which is 6% lower than that of pure natural rubber. As previously discussed, the crosslink density of the vulcanizate containing 2 phr of 3F-PBu-3F is 24.32% higher than that of pure NR. The higher crosslink density effectively suppresses the diffusion of oil into the vulcanizates. Furthermore, the main chain repeating structural unit of polybutadiene exhibits greater oil resistance compared to that of NR [36]. The introduction of fluorine atoms into the network also contributes to the enhancement in oil resistance [37,38]. In summary, the incorporation of 3F-PBu-3F significantly improves the oil resistance of the vulcanizates.

Table 1 lists the effects of different modification systems on the oil resistance of natural rubber. Among them, epoxidation and fluorination modifications are carried out through physical blending, with an additive amount of more than 20% to achieve the best oil resistance. In contrast, the fluorine-terminated modification method used in this study has an additive amount of only 1–5%, resulting in relatively weaker oil resistance. It is worth noting that chlorination and nanocomposite modification are achieved through chemical modification followed by physical blending. However, their additive amounts are also relatively low, resulting in a similar swelling rate in this study.

### 2.5. Mechanical and Thermal Stability Properties of NR/3F-PBu-3F Vulcanizates

The mechanical properties, including tensile strength and elongation at break, of the NR/3F-PBu-3F vulcanizates are presented in Figure 9. The tensile strength of the NR/3F-PBu-3F vulcanizates was found to decrease nonlinearly from 17.41 MPa to 10.37 MPa as the 3F-PBu-3F content increased from 0 phr to 5 phr. Of them, the maximum tensile strength of 17.56 MPa was achieved when the content of 3F-PBu-3F was 1 phr, which showed a slight improvement in tensile strength compared with NR. Meanwhile, the elongation at break initially decreased from 866.6% to 807.9% and then increased to 873.9%. In this sense, a shift from a stiffer material to one with more flexibility was observed at higher 3F-PBu-3F levels. Firstly, the relatively low molecular weight of the liquid 3F-PBu-3F led to its inferior strength. Generally, even high-molecular-weight polybutadiene exhibits a lower tensile strength compared to natural rubber. To confirm this, the mechanical properties of NR containing 1–5 phr HO-PBu-OH were also evaluated in terms of tensile strength and elongation at break. As shown in Figure 9, similar trends can be detected. Additionally, the tensile strength of NR/3F-PBu-3F was consistently lower than that of NR/HO-PBu-OH. Given that polymers with heterogeneous structures (as reported in NR/HO-PBu-OH blends) generally exhibit higher tensile strength than those with homogeneous structures, this observation is quite reasonable. Secondly, an increase in crosslink density can enhance the strength of the vulcanizates to a certain extent. When the content of the liquid 3F-PBu-3F would be further increased, the slower crosslinking rate under sulfur vulcanization conditions may result in a certain amount of uncrosslinked 3F-PBu-3F. The presence of this uncrosslinked 3F-PBu-3F, acting as a plasticizer, can cause the observed decrease in tensile strength and the increase in elongation at break. Based on the mechanical properties, a threshold content of 3F-PBu-3F can be 1 phr.

As shown in Figure 10 and Figure 11, the thermal stability of the vulcanizates can generally be improved with the incorporation of 3F-PBu-3F. Two specific temperatures, including TIDT (5% weight loss) and Tmax (the maximum rate of weight loss), are chosen to evaluate the thermal stability of the vulcanizates. It can be seen that TIDT increases by 7 °C with the incorporation of 1 phr 3F-PBu-3F. Furthermore, the Tmax of the vulcanizate containing 1 phr 3F-PBu-3F increases significantly from 378 °C to 384 °C. It should be pointed out that hydroxyl-terminated polybutadiene has poor thermal stability, typically showing thermal decomposition at 150–250 °C following a first-order law [43]. However, polymers with the incorporation of the fluorine element generally show improved thermal stability [44,45]. Therefore, by modifying the hydroxyl-terminated groups of polybutadiene with fluorine atoms and then introducing the modified polybutadiene (3F-PBu-3F) into NR, the thermal stability of the vulcanizates can be significantly improved.

## 3. Materials and Methods

### 3.1. Materials

Natural rubber (NR, Grade V) was supplied by Hainan Natural Rubber Industry Group Co., Ltd. (Haikou, Hainan, China). Hydroxyl-terminated polybutadiene (HTPB, *M*_n_ = 2800) was purchased from Meryer Co., Ltd. (Shanghai, China). Thionyl chloride (AR) and azobis(isobutyronitrile) (AIBN and AR) were purchased from Aladdin (Shanghai, China). 2,2,2-Trifluoroethyl acrylate (AR), tetrahydrofuran (THF and AR), toluene (AR), *n*-hexane (AR), and methanol (AR) were all obtained from Xilong Scientific (Guangzhou China). Mercaptopropionic acid (AR and TGA), zinc oxide (AR), stearic acid (SA and AR), and 2-mercaptobenzothiazole (MBT and AR) were purchased from Macklin Co., Ltd. (Shanghai, China). Sulfur (S, industrial grade) was supplied by Qingdao LuChuan Chemical Co., Ltd. (Qingdao, China). Deionized water was prepared using an ultrapure water machine in our laboratory.

### 3.2. Methods

#### 3.2.1. Synthesis of Fluorine-Terminated Polybutadiene (3F-PBu-3F)

The synthesis routine of fluorine-terminated polybutadiene (3F-PBu-3F) is illustrated in Figure 12. First, 40 mL of 2,2,2-trifluoropropyl acrylate and 27.5 mL (310.15 mmol) of mercaptopropionic acid were dissolved in 50 mL of THF. Subsequently, 0.05 g (0.304 mmol) of AIBN was added to initiate a thiol-ene radical reaction. After reacting at 60 °C for 12 h, the THF was removed via rotary evaporation at room temperature. The obtained 30 g of 4-((3-Oxo-3-(2,2,2-trifluoroethoxy)propyl)thio)butanoic acid as the product was recrystallized from n-hexane and subsequently dried in an oven at 40 °C for 24 h to obtain 4-((3-Oxo-3-(2,2,2-trifluoroethoxy)propyl)thio)butanoic acid.

Fluorine-terminated polybutadiene was synthesized through the substitution reaction between α, ω-dihydroxy-terminated polybutadiene (HO-PBU-OH) and fluorine-containing polymers in this work. First, HO-PBU-OH was dried via azeotropic distillation with anhydrous toluene for preparation. Second, 30 g of fluorine-containing polymers was introduced to a round-bottom flask equipped with a magnetic stirring bar. The flask was then sealed with vacuum grease, and 45 mL of anhydrous toluene was injected to dissolve the polymers. To avoid a vigorous exothermic reaction, the flask was placed in an ice water bath environment, and then 20 mL (275.36 mmol) of thionyl chloride was injected. The acyl chloride reaction was carried out for 24 h. Afterward, residual thionyl chloride was removed through distillation at room temperature for 4 h followed by distillation in vacuo for 6 h. Finally, the prepared anhydrous HO-PBU-OH was injected into the system, and the substitution reaction was carried out at 0 °C for 24 h. The reacted mixture was added to 500 mL of methanol to obtain precipitates. The product, 3F-PBu-3F, was purified using this dissolution-precipitation procedure five times and then dried under vacuum at 25 °C for 24 h.

#### 3.2.2. Preparation of NR/3F-PBu-3F Vulcanizates

Prior to compounding, NR was masticated in an open-roll mixing mill at room temperature for 10 min. NR-based blends with different amounts of 3F-PBu-3F were prepared according to GB/T 6038-2006. Table 2 presents the amount ratios of NR to 3F-PBu-3F. Subsequently, the blends were cured in a vulcanizing press machine. The curing conditions, including the determination of the optimum cure time (*t*_C90_), were determined using a moving die rheometer in accordance with GB/T 16584-1996 at 145 °C.

### 3.3. Characterization and Measurement

#### 3.3.1. Nuclear Magnetic Resonance Spectroscopy (^1^H NMR)

Each sample was dissolved in deuterated chloroform, and nuclear magnetic resonance spectra were recorded on a Bruker AVANCE NEO 400 MHz NMR spectrometer (Bruker BioSpin AG, Basel, Switzerland) with tetramethylsilane (TMS) as the internal reference.

#### 3.3.2. Fourier Transform Infrared Spectroscopy (FT-IR)

FT-IR analysis was performed using a Perkin-Elmer Spectrum One FT-IR spectrometer (Waltham, MA, USA). The NR/3F-PBu-3F vulcanizates were measured using Attenuated Total Reflectance (ATR). Other samples were dissolved in dichloromethane, dropped onto a KBr pellet, and then dried under infrared light, respectively. After the residual solvent was evaporated completely, each sample was measured. For all samples, a wavelength range of 4000 cm^−1^ to 600 cm^−1^ was used, with 32 scans conducted at a resolution of 8 cm^−1^.

#### 3.3.3. Scanning Electron Microscopy (SEM)

The morphologies of the vulcanizates were observed using a Gemini300 (Carl Zeiss AG, Baden-Württemberg, Germany) scanning electron microscope at an acceleration voltage of 3 kV. Prior to observation, all vulcanizates were cryo-fractured in liquid nitrogen, and the fracture surfaces of the vulcanizates were then coated with gold.

#### 3.3.4. Oil Resistance and Crosslink Density

The oil resistance and crosslink density of the NR/3F-PBu-3F vulcanizates were measured according to ASTM D471 and GB/T 1690-2010, respectively. Rectangular samples with a size of 20 mm × 20 mm × 1 mm were immersed in IMR903 type engine oil. After the target time, residual oil on the surface of the samples was tapped with filter paper, and the mass of the samples was weighted. The Mass Change Rate of the samples was calculated according to Equation (1):(1)$$\mathrm{Mass}\mathrm{Change}\mathrm{Rate}\;\left(\%\right)=\frac{\left(m_{i}-m_{i}\right)}{m_{i}}\times 100\%$$
where *m*_0_ is the initial mass of the sample, and *m_i_* is the mass of the sample after immersion for different durations.

Likewise, the crosslink density of the samples was calculated according to the Flory–Rehner equation (Equation (2)) below. For the purpose of swelling equilibrium, the mass difference of the last two samples was kept at less than 0.01 g.(2)$$M_{c}=-\rho_{2}V_{1}\varphi_{2}^{\frac{1}{3}}/[ln(1-\varphi_{2})+\varphi_{2}+{ℵ}_{1}\varphi_{2}^{2}]$$
where *M_c_* is the average molecular weight between crosslinks, which represents the average molecular weight of the polymer chains between two adjacent crosslinking points.

#### 3.3.5. Mechanical Properties

The tensile strength of the NR/3F-PBu-3F vulcanizates was measured on a universal testing machine (GOTECH AI-3000). Dumbbell-shaped specimens with a size of 75 mm × 4 mm × 1 mm were tested at a strain rate of 500 mm/min at 25 °C.

#### 3.3.6. Thermogravimetric Analysis (TGA)

Thermogravimetric analysis (TGA) was performed using a Mettler Toledo TGA/DSC 1/1100 thermogravimetric analyzer (Switzerland). Typically, 10 mg of sample was heated from 25 °C to 700 °C with a ramping rate of 10 °C/min under a nitrogen atmosphere.

## 4. Conclusions

In this paper, the synthesis of 4-((3-Oxo-3-(2,2,2-trifluoroethoxy)propyl)thio)butanoic acid and the fluorination modification of hydroxyl-terminated polybutadiene were successfully achieved. The fluorine-modified polybutadiene (3F-PBu-3F) was incorporated into natural rubber, followed by curing with sulfur to prepare NR/3F-PBu-3F vulcanizates. The NR/3F-PBu-3F vulcanizates exhibited improved oil resistance and thermal stability compared with natural rubber. In addition, the relationship between the crosslink density and oil resistance and thermal stability was discussed, and the increase in crosslinking density has a positive effect on vulcanizates. Moreover, according to SEM imaging, homogeneous morphologies of the vulcanizates were obtained, which are remarkably different from those of the NR/polybutadiene vulcanizates reported previously. This was owing to the improvement in miscibility, which was discussed regarding the low molecular weight of liquid 3F-PBu-3F to improve diffusion capability, as well as specific hydrogen bonding interactions between proteins and the fluorine atoms of 3F-PBu-3F. Consequently, we acquired NR/3F-PBu-3F vulcanizates to offer a valuable solution for improving the stability in hyperthermy and oil-containing environments. It should be pointed out that the cost efficiency and environmental friendliness of this strategy are not very competitive due to the use of TFEA. Nevertheless, this work provides insights for enhancing the oil resistance and thermal stability of NR and lays a foundation for future exploration.

## Figures and Tables

**Figure 1 ijms-26-03410-f001:**
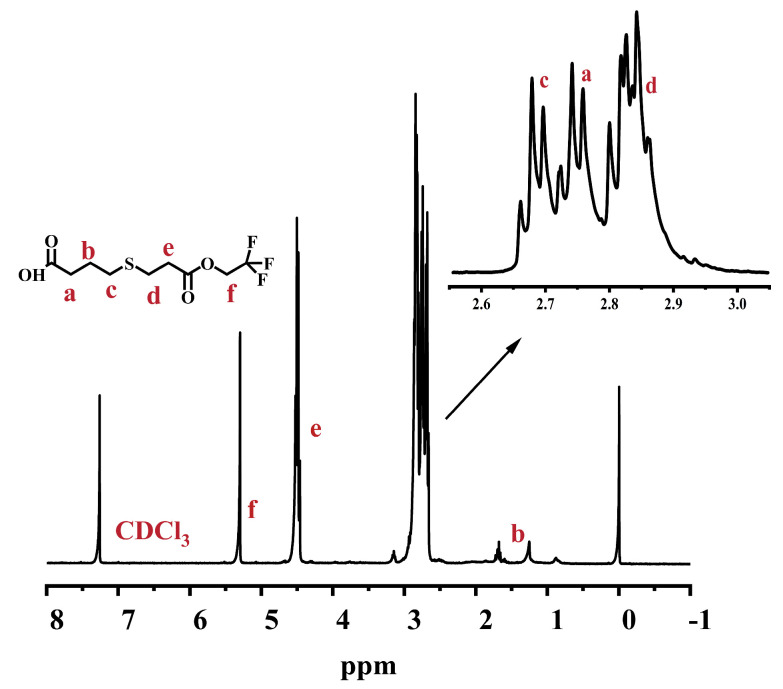
^1^H-NMR spectrum of 4-((3-Oxo-3-(2,2,2-trifluoroethoxy)propyl)thio)butanoic acid, where -CH_2_- group is in α position of -CF_3_ at 5.53 ppm.

**Figure 2 ijms-26-03410-f002:**
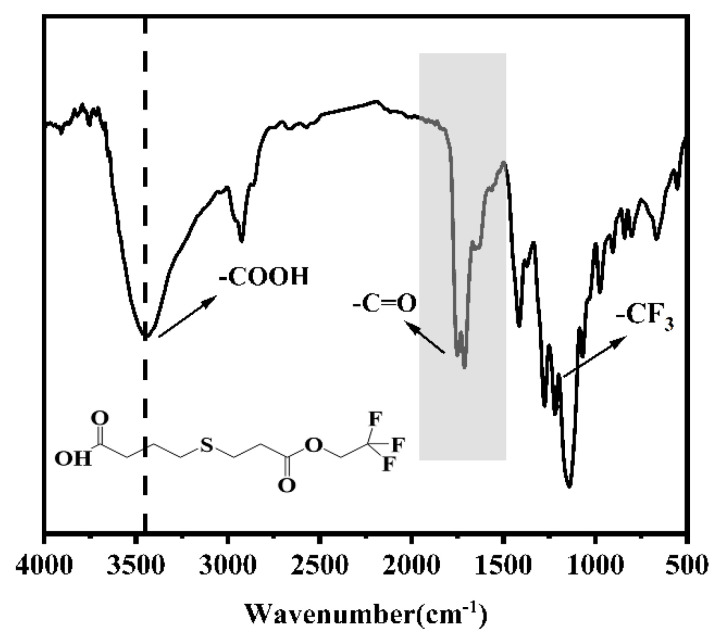
The infrared spectrum of 4-((3-Oxo-3-(2,2,2-trifluoroethoxy)propyl)thio)butanoic acid, where the -CF_3_ characteristic signal is at 1280 cm^−1^.

**Figure 3 ijms-26-03410-f003:**
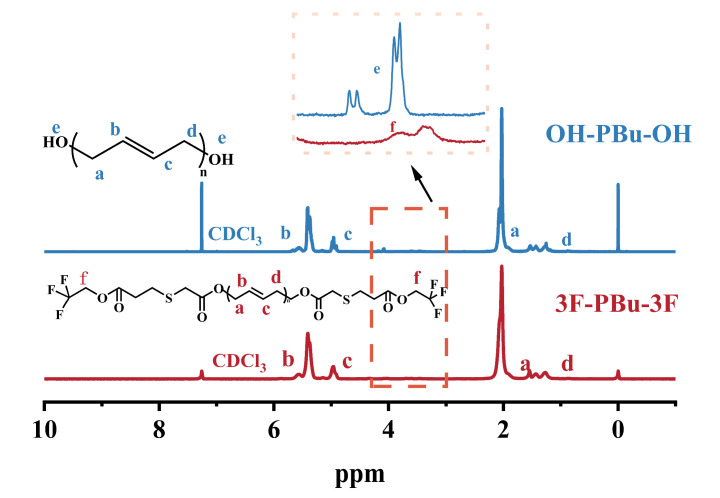
Comparison of ^1^H NMR spectra of OH-PBu-OH and 3F-PBu-3F. Blue trace: OH-PBu-OH; red trace: 3F-PBu-3F.

**Figure 4 ijms-26-03410-f004:**
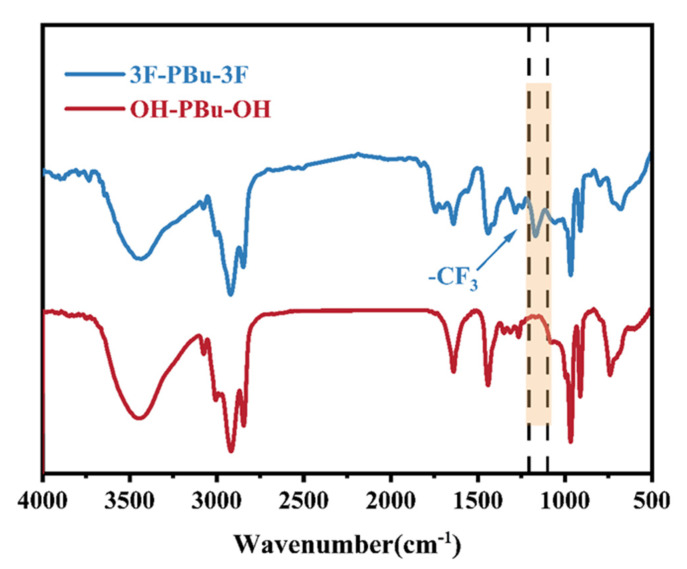
Comparison of FT-IR spectra of OH-PBu-OH and 3F-PBu-3F. Blue trace: OH-PBu-OH; red trace: 3F-PBu-3F.

**Figure 5 ijms-26-03410-f005:**
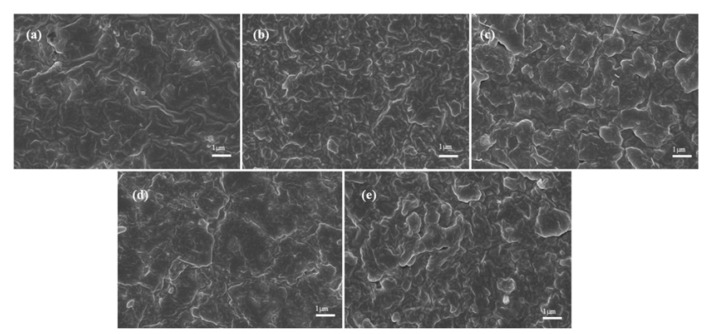
SEM images of vulcanizates with different contents of 3F-PBu-3F ((**a**) 1 phr, (**b**) 2 phr, (**c**) 3 phr, (**d**) 4 phr, and (**e**) 5 phr).

**Figure 6 ijms-26-03410-f006:**
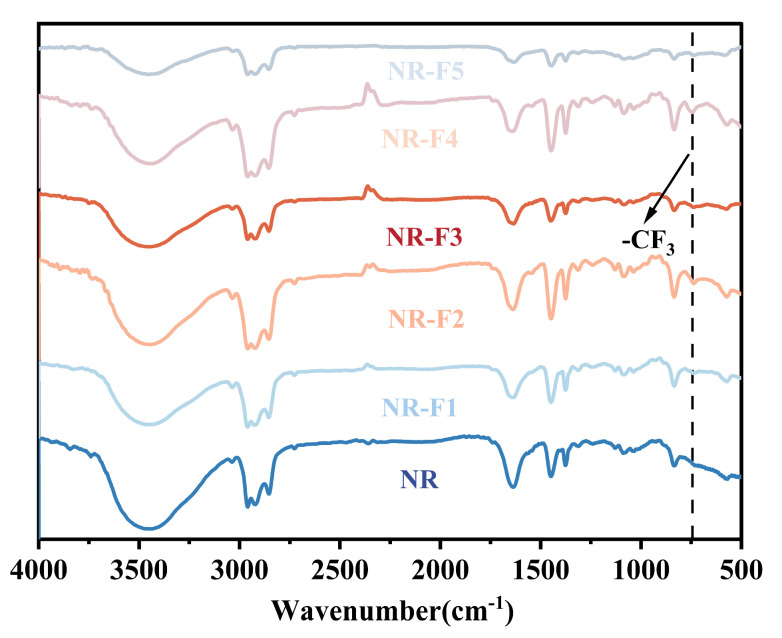
FT-IR spectra of NR/3F-PBu-3F vulcanizates.

**Figure 7 ijms-26-03410-f007:**
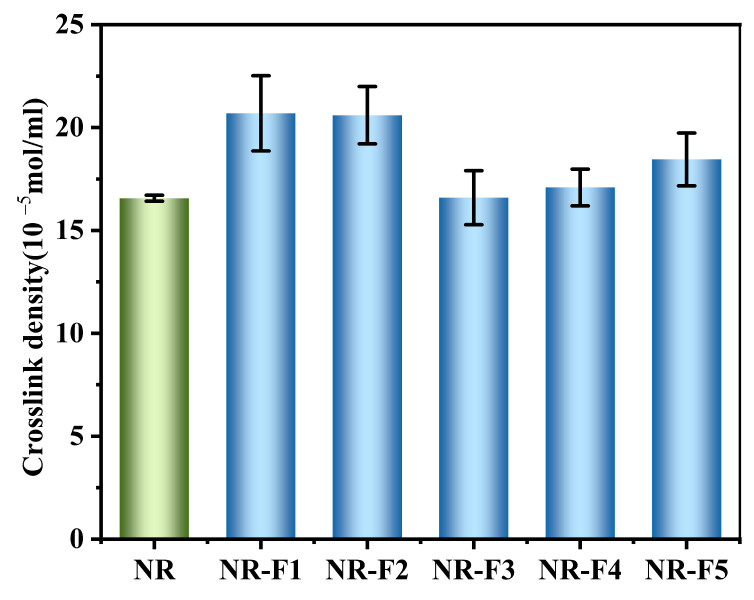
Crosslink density of NR/3F-PBu-3F vulcanizates.

**Figure 8 ijms-26-03410-f008:**
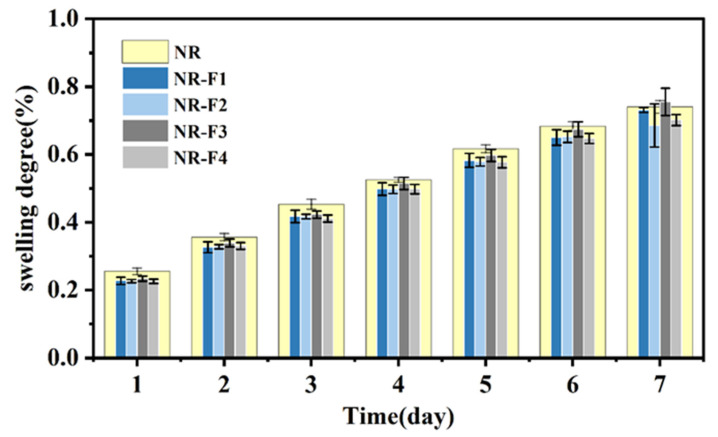
Oil resistance of NR/3F-PBu-3F vulcanizates.

**Figure 9 ijms-26-03410-f009:**
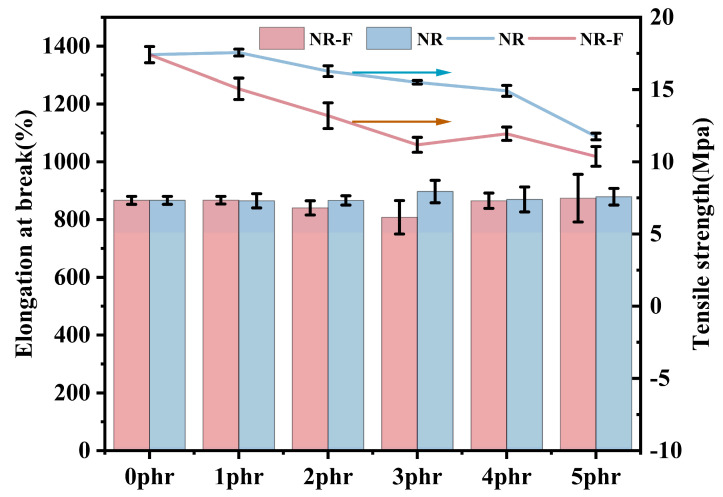
Mechanical properties of NR/3F-PBu-3F vulcanizates.

**Figure 10 ijms-26-03410-f010:**
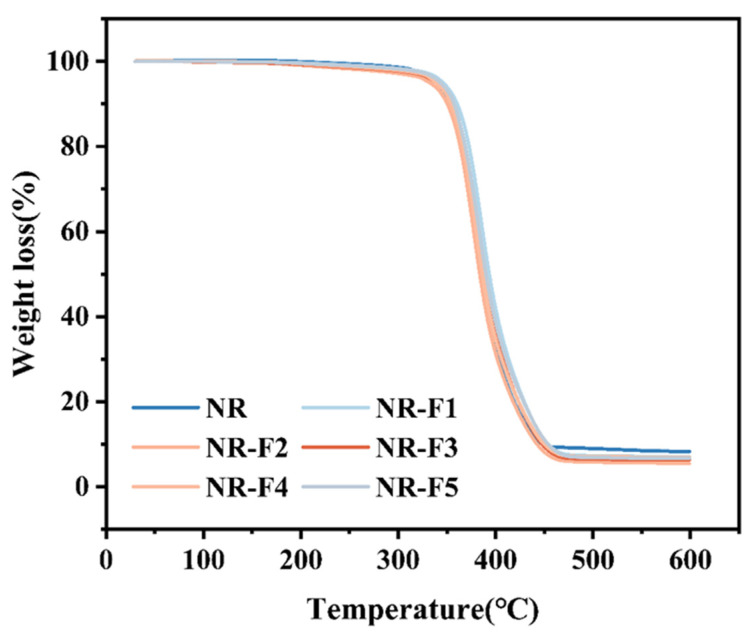
TGA plots of NR/3F-PBu-3F vulcanizates.

**Figure 11 ijms-26-03410-f011:**
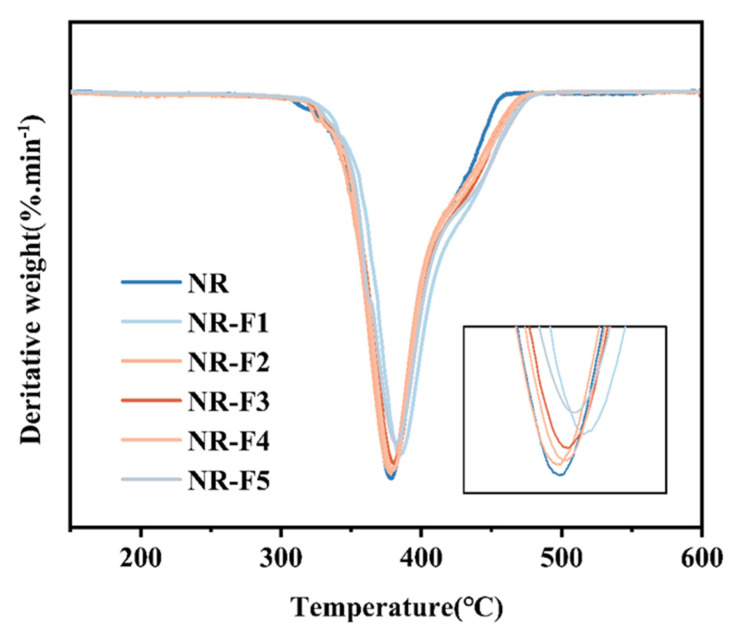
Differential thermal analysis of NR/3F-PBu-3F vulcanizates.

**Figure 12 ijms-26-03410-f012:**
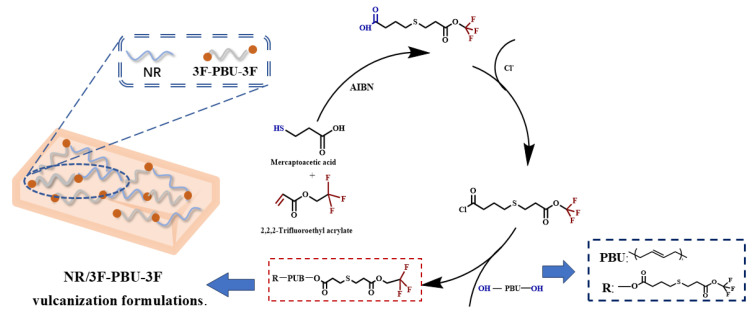
Synthesis routine of 3F-PBu-3F.

**Table 1 ijms-26-03410-t001:** Formula of NR/3F-PBu-3F vulcanizates. Specimens were immersed in IRM 903 reference oil at 23 ± 2 °C for 72 h according to ASTM D471.

Modification Method	Material System	Mass Change Rate (%)	Reference
Fluorinated end-group	NR/3F-PBu-3F	40–45	This work
Epoxidation	ENR	10–30	[39]
Fluorinated	PVDF/MGNR	15–25	[40]
Chlorinated	PVC/NRF	30–40	[41]
Nanocomposite	ENR40/VAE/nSiO_2_	30–80	[42]

**Table 2 ijms-26-03410-t002:** Formula of NR/3F-PBu-3F vulcanizates.

Sample	NR (phr)	3F-PBu-3F (phr)	ZnO (phr)	SA (phr)	MBT (phr)	S (phr)
S0	100	0	6	0.5	0.5	3.5
S1	100	1	6	0.5	0.5	3.5
S2	100	2	6	0.5	0.5	3.5
S3	100	3	6	0.5	0.5	3.5
S4	100	4	6	0.5	0.5	3.5
S5	100	5	6	0.5	0.5	3.5

## Data Availability

The data that support the findings of this study are available from the corresponding author upon reasonable request.

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
