# Peer review of "Fluorine-Terminated Liquid Polybutadiene: A Novel Approach to Enhancing Oil Resistance and Thermal Stability in Natural Rubber"

_ijms, 2025, doi:10.3390/ijms26073410_

Round 1
Reviewer 1 Report
Comments and Suggestions for Authors
The subject of the manuscript focused on fluorine-terminated liquid polybutadiene: a novel approach to enhancing oil resistance and thermal stability in Natural Rubber is in good relevance with the scope of International Journal of Molecular Sciences.
The manuscript addresses the problem of improving the properties of natural rubber by incorporation of fluorine-terminated liquid polybutadiene. The authors present the results achieved for fluorine-terminated liquid polybutadiene showing the effect of the modification on oil resistance, mechanical properties, and thermal stability of the obtained vulcanizates. The manuscript requires the following corrections and additions:
- Lack of sufficient discussion of the limitations of the study. The manuscript presents positive results but does not discuss potential drawbacks of the approach. For example, there is no mention of the effects of fluorinated modification on production costs and the environment.
- Poor analysis of mechanical properties of NR/3F-PBu-3F. The manuscript mentions a decrease in tensile strength with increasing the content of 3F-PBu-3F, but there is no in-depth analysis of this phenomenon. The influence of the degree of cross-linking on elasticity and fracture toughness could be explained in more detail.
- Limited comparison with existing solutions. The work focuses on the obtained results, but does not directly compare them with other methods of improving oil and temperature resistance, e.g. using nanofillers or other modifiers.
- Lack of aging tests. It is not known how stable the material properties are over time, especially under long-term exposure to high temperatures and oils.
- In several places the manuscript contains sentences that are difficult to understand, which may make it difficult to receive the content. An example is "Combining the with the analysis of 1H NMR and FT-IR discussed above, it can be concluded that..." - this sentence is ungrammatical.
The English must be improved to more clearly express the research.
Reviewer 2 Report
Comments and Suggestions for Authors
General Comments:
The manuscript presents an interesting study on the fluorination modification of hydroxyl-terminated polybutadiene and its incorporation into natural rubber. The research is relevant to improving the oil resistance and thermal stability of natural rubber, which is crucial for industrial applications. The methodology appears well-structured, and the findings provide valuable insights into material performance. However, some areas require clarification, revision, and grammatical improvements to enhance readability and scientific rigor.
-
Grammar and Readability Improvements
- Line 40: “Fluorine has high electronegativity (3.98) and small atomic radius (0.67 Å), which makes the carbon-fluorine bond stably...” → should be rewritten as "Fluorine has a high electronegativity (3.98) and a small atomic radius (0.67 Å), which makes the carbon-fluorine bond highly stable."
- Line 67: "The signal of methylene protons (–CHâ‚‚–) connected to the ester group (–COO–) at 4.95 ppm, attributing to electron-withdrawing effect of sulfur." → should be rewritten as "The signal of methylene protons (–CHâ‚‚–) connected to the ester group (–COO–) appears at 4.95 ppm, which is attributed to the electron-withdrawing effect of sulfur."
-
Formatting Issues
- Figures 1-4: Ensure that all figures are properly labeled, and their captions fully describe the contents. Some figure legends are too brief.
- Table 1 (Line 251): The table should include standard deviation values where applicable to provide statistical validity to the data.
-
References and Citations
- Lines 321-406 (References Section): Some references are incomplete, and their formatting does not adhere to the journal’s requirements. Ensure that all references are consistently formatted.
- Specific Comments:
- Lines 8-16 (Abstract): The abstract should clearly state the hypothesis of the study. While the improvements in oil resistance and thermal stability are discussed, the novelty of the approach should be emphasized more explicitly.
- Lines 19-39 (Introduction): The introduction provides a strong background on natural rubber limitations. However, the transition to the study’s specific objectives needs more clarity. Consider adding a clear research hypothesis or objective statement towards the end of the introduction.
- Lines 54-61: More details are needed regarding the synthesis of fluorine-terminated polybutadiene (3F-PBu-3F). What were the reaction conditions (e.g., temperature, reaction time) that led to the successful modification?
- Lines 64-72 (NMR Analysis): The description of the NMR results is quite technical. While this is necessary, a clearer explanation of how these findings confirm the success of fluorination would benefit readers unfamiliar with NMR interpretation.
- Lines 77-85 (FT-IR Analysis): The statement "Combining the with the analysis of 1H NMR and FT-IR discussed above..." is unclear. Consider rewording to explicitly state how these analyses support the conclusions.
- Lines 145-174 (Crosslink Density & Oil Resistance): The relationship between crosslink density and oil resistance is interesting. However, the discussion should incorporate more comparative literature references to contextualize the improvements. How do these results compare to other modifications aimed at improving oil resistance?
- Lines 180-189 (Mechanical Properties): The reason behind the decrease in tensile strength with increasing 3F-PBu-3F content should be discussed in more depth. Is there a threshold beyond which the incorporation of fluorinated polybutadiene weakens the rubber matrix?
- Lines 207-241 (Materials and Methods): The synthesis method should specify purification steps more clearly. How was the removal of unreacted materials ensured?
Round 2
Reviewer 1 Report
Comments and Suggestions for Authors
The subject of the manuscript focused on fluorine-terminated liquid polybutadiene: a novel approach to enhancing oil resistance and thermal stability in Natural Rubber is in good relevance with the scope of International Journal of Molecular Sciences.
The manuscript addresses the problem of improving the properties of natural rubber by incorporation of fluorine-terminated liquid polybutadiene. The authors present the results achieved for fluorine-terminated liquid polybutadiene showing the effect of the modification on oil resistance, mechanical properties, and thermal stability of the obtained vulcanizates.
Comments on the Quality of English LanguageThe English could be improved to more clearly express the research